# Multi-Core Fiber Bragg Grating and Its Sensing Application

**DOI:** 10.3390/s24144532

**Published:** 2024-07-13

**Authors:** Xiaotong Zhang, Hongye Wang, Tingting Yuan, Libo Yuan

**Affiliations:** 1Center for Advanced Manufacturing and Future Industry, Future Technology School, Shenzhen Technology University, Shenzhen 518118, China; zhangxiaotong@sztu.edu.cn; 2Key Lab of In-Fiber Integrated Optics, Ministry Education of China, Harbin Engineering University, Harbin 150001, China; wanghongye92@hrbeu.edu.cn; 3Photonics Research Center, School of Optoelectronic Engineering, Guilin University of Electronic Technology, Guilin 541004, China

**Keywords:** multi-core fiber, fiber Bragg grating, optical fiber sensor, application

## Abstract

With the increase in the demand for large-capacity optical communication capacity, multi-core optical fiber (MCF) communication technology has developed, and both the types of MCFs and related devices have become increasingly mature. The application of MCFs in the field of sensing has also received more and more attention, among which MCF fiber Bragg grating (FBG) devices have received more and more attention and have been widely used in various fields. In this paper, the main writing methods of MCF FBGs and their sensing applications are reviewed. The future development of the MCF FBG is also prospected.

## 1. Introduction

To cope with the huge challenge of the explosive growth of information to the capacity demand of optical communication systems, the development of multi-core fiber (MCF) has gradually emerged as an important solution to the capacity crisis [1,2,3]. MCF refers to optical fibers with multiple cores within the same cladding, which can provide multiple independent spatial channels in a single optical fiber. In recent years, with the continuous improvement of technology, the problem of inter-core cross-talk that hinders the increase in core density has been solved, the number of cores of MCF has gradually increased, and remote optical communication transmission has been achieved [4,5]. Space division multiplexing technology based on MCF is the main method to solve the current capacity crisis. Therefore, the continuous development of MCF technology has attracted more and more attention. In addition to its application in the field of communication, MCFs are also being gradually applied to various fields, including new optical fiber sensors [6,7], microwave photonic devices [8,9], and laser systems with coherent beam combinations [10,11].

Since Hill et al. first developed the world’s first permanent optical fiber grating—fiber Bragg grating (FBG)—in germanium-doped optical fibers using the standing wave method in 1978 [12], this new technology of fabricating gratings in optical fibers has attracted great interest. This is because the emergence of fiber gratings has not only brought another milestone revolution to optical fiber communication and optical fiber sensing but also enabled people to design and fabricate a large number of new optical active/passive devices and sensors based on FBG [13,14,15]. With the gradual maturity of preparation technology for special-structured optical fibers such as MCF, researchers have researched the application of MCF FBGs. In terms of communication, MCF FBGs are an effective spatial division multiplexing technique [16], while FBGs inscribed on few-mode fibers can achieve mode division multiplexing/demultiplexing [17,18,19]. Therefore, the use of multi-core few-mode fiber gratings can simultaneously achieve multiple multiplexing techniques, greatly expanding transmission capacity. In the field of life sciences, Raman spectroscopy is used to find pathological tissues, and a commonly used method is to use optical fibers to receive probe light and use FBGs to suppress elastically scattered light. However, the small core diameter of traditional single-mode fiber (SMF) and low light collection efficiency hinder the development of optical fiber probes. The development of MCF FBGs has effectively solved this problem, making optical fiber Raman probes more widely applicable [20]. By utilizing the filtering characteristics of FBGs, the influence of hydroxyl groups can be effectively suppressed [21], promoting the development of photonic lanterns. The use of MCF FBGs in the preparation of photonic lanterns not only increases the number of ports but also ensures the compactness of the overall structure, which is of great significance in astronomical photonics [22]. MCF FBGs have also made significant progress in the field of sensing. MCF FBGs can measure three-dimensional bending while eliminating the influence of temperature, enabling multi-parameter measurement. Therefore, inscribing the FBG arrays on MCF can achieve quasi-distributed three-dimensional shape sensing [23,24], which has a wide range of applications in medical [25], aviation [26], robotics [27], and other fields.

At present, whether it is the inscribing method [28,29,30,31], theoretical research, or application of MCF FBGs, they have achieved rapid development. In this article, we review the main preparation methods of MCF FBGs, introduce the current sensing applications of multi-core fiber gratings, and then discuss the challenges faced by MCF FBGs and their future predictions. The last chapter gives a summary of the whole article.

## 2. Methods for Inscribing Bragg Gratings in MCF

MCFs have been developed in a variety of forms. Among them, to avoid high cross-talk between adjacent cores, the core spacing is set relatively large (generally greater than 30 μm) [32]. This type of MCF is called a weakly coupled MCF, and the optical signal can be transmitted independently in cores at different positions. The gratings inscribed in the weakly coupled MCF still need to meet independent transmission requirements.

Since Gander et al. first used the ultraviolet exposure method to inscribe the FBG on four-core fiber in 2000 [33], the technology of inscribing MCF FBGs has become increasingly mature. Due to the geometric arrangement of the cores in MCF, one of the main challenges in inscribing FBGs on MCF is how to uniformly imprint FBGs on each core or the selected cores. Currently, the inscribing methods for MCF can be divided into two types based on the number of cores being inscribed simultaneously: one-time inscribing (full-core inscribing) and selective core inscribing (core-by-core inscribing). Full-core inscribing involves simultaneously imprinting FBGs with similar or identical characteristics in all cores of the MCF, while selective inscribing involves selecting specific cores in the MCF for FBG inscribing. Full-core inscription often needs to address the non-uniformity issues arising from the lens effect of the optical fiber and the shadowing effect of the cores. Therefore, many methods are modified based on the traditional phase mask method to achieve good uniformity of the FBGs in different cores. The core-by-core inscription method allows for separate inscription on each core, with the inscription beam being focused individually on each core. This can effectively mitigate the lensing effect and is not constrained by the phase mask, resulting in higher inscription accuracy. It also enables the fabrication of gratings on complex-structured multi-core fibers.

### 2.1. Full-Core Inscribing FBGs

Affected by the structure of the fiber itself, its cylindrical structure is equivalent to a lens. When the light is irradiated to the fiber from the side, it will converge, resulting in different light intensity on each core of the MCF, resulting in the different reflectivity of FBG on each core, which is the main reason for affecting the inscribing quality of the MCF FBG. The main ideas of inscribing FBGs on the whole cores of MCF to achieve the same FBG parameters on each core are divided into adjusting the structure of the fiber itself to offset the different irradiated intensity of each core caused by the lens effect or modulating the incident light field so that the light field irradiated on each core is more uniform, so that the FBG on each core has similar resonant wavelength and reflectivity.

As early as 2007, Askin proposed adjusting the positions and refractive indices of the four cores of a four-core fiber [34]. Based on the intensity distribution of ultraviolet light irradiated on the fiber, the doping concentration, core positions, and core spacing of the four cores were adjusted so that the refractive index changes in each core were relatively consistent when ultraviolet light was irradiated on the fiber. Finally, a multi-core fiber grating with good consistency was obtained. Professor Yuan Libo proposed a method to make the exposure rates of each core more consistent by changing the arrangement of the cores of the multi-core fiber [35], as shown in Figure 1a. In addition to the above MCF FBG with good consistency obtained by adjusting the fiber cores structure, Emma Lindley et al. proposed using a side-polished quartz tube to offset the lens effect caused by the fiber [36,37]. The inner diameter of the quartz capillary is about equal to the diameter of the fiber, so the light incident on the fiber core from the polished surface of the quartz tube can be regarded as flat light, as shown in Figure 1b. The seven-core fiber (SCF) FBG inscribed by this method has high wavelength consistency. Deng Shunge et al. proposed a rotating inscribing method for MCF FBG [38]. A pair of rotators rotated the fiber several times during inscribing so that the incident light could be irradiated to the core of the MCF from multiple directions. By reasonably selecting the single inscribing time and rotation times, this method could obtain reflected signals with low bandwidth, high reflection, and good uniformity.

In addition, Professor Yuan used a one-time grating writing method with defocusing to study the effect of different defocusing distances on the quality of the grating [39]. Finally, a multi-core fiber grating with a center wavelength error less than 0.783 nm was successfully prepared on a four-core fiber distributed in a triangular shape, as shown in Figure 1c. Leixin Meng et al. proposed the use of external ultraviolet transparent media to eliminate the lensing effect caused by the large difference in refractive index between air and fiber [40]. Potassium chloride solution is used as the matching liquid, and the refractive index of the liquid is controlled by adjusting the concentration of potassium chloride, so that the refractive index of the configured liquid is the same as that of the fiber cladding, as shown in Figure 1d. In this way, the team inscribed the SCF FBG array with high consistency.

In addition to the above methods, the holographic interference method (HIM) is also a widely used method in recent years. HIM utilizes the interference fringes produced by the interference of two beams to inscribe gratings in the fiber core. Its characteristic is that the reflection mirror of the interferometer can be rotated to change the period of the interference pattern, thereby altering the period of the inscribed gratings. Among the reported HIMs, most are achieved using the Talbot interferometer HIM. Compared to the phase mask method, this method offers more flexibility. With a sufficiently strong light source, the FBGs on all cores can be achieved in a single exposure, making the setup simple and effective.

Stępień et al. utilized a standard Talbot interferometer to inscribe uniform FBGs in all cores of a seven-core fiber in a single exposure, achieving good uniformity of the Bragg wavelengths [41]. In 2016, Hoe et al. combined fiber drawing with the HIM to inscribe FBGs in all cores of a four-core fiber in a single exposure. The FBGs produced by this technique are referred to as Draw Tower Grating (DTG) [42], and the inscription setup is shown in Figure 2a. This method allows for the direct fabrication of FBGs during the fiber drawing process, offering advantages for the mass customization of gratings. In 2018, Wang et al. used a modified Talbot interferometer combined with a lens vertical scanning method to simultaneously write FBGs in a seven-core fiber. By employing a beam scanning approach, they ensured adequate exposure for each core, with the Bragg wavelengths of the individual cores differing by no more than 1.24 nm, as shown in Figure 2b [43].

#### 2.1.1. UV Laser Writing Method

Bao Weijia from Aston University proposed a method that allows for core-by-core writing based on the phase mask [44]. Firstly, the position of the fiber core is determined by using a microscopic observation device. Then, the grating writing quality of different beam-focusing positions is studied to determine the beam-focusing position. Finally, by adjusting the distance between the cylindrical lens and the fiber, the fiber core to be written with the grating is placed at the focus of the light, and the grating writing is realized for the four-core fiber core by core, as shown in Figure 3a. In addition, by placing the laser focus at the midpoint of the connecting line of any two fiber cores, the selective writing of dual-core can be achieved. By adjusting the position of the laser intersection point and placing the laser focus in the middle of the connection line between the two cores, it is possible to achieve the selective writing of both cores simultaneously. In this case, the reflection intensity of the FBG in the core closer to the incident beam will be slightly higher than that of the FBG in the remote core, indicating that the core irradiated by the laser first will be exposed to stronger influence. 

Jantzen et al. from the University of Southampton proposed a small spot ultraviolet direct writing method based on double-beam interference [45]. This method splits the laser beam into two before focusing and then converges the two beams into an interference spot with a diameter of 14 μm on one core of the optical fiber. The interference spot of this size can completely cover the core, and then, the optical fiber is moved to achieve the desired length of grating writing, as shown in Figure 3b. This method has high flexibility, where the period of the grating can be determined by the angle between the two interfering beams, and the length of the grating can be determined by the length of the optical fiber movement.

#### 2.1.2. Femtosecond Laser Writing Gratings

In recent years, with the development of laser technology, the use of high-energy femtosecond lasers as a light source has become one of the important schemes for selectively writing gratings in multi-core fibers [46,47]. Femtosecond lasers have very high peak power and extremely short pulse widths, which can produce nonlinear effects (nonlinear field ionization and avalanche ionization) when interacting with transparent media, thereby permanently changing the refractive index of the material. Due to the different writing mechanisms, femtosecond laser writing has higher flexibility compared to ultraviolet exposure methods. The fibers used do not need to be photosensitive, so there is no need for hydrogen loading treatment of the fibers, and the fiber coating does not need to be removed. The gratings also have some unique properties, such as high-temperature resistance. The preparation methods of fiber gratings based on femtosecond lasers are divided into two approaches: one is direct writing with femtosecond lasers, and the other is grating writing assisted by phase templates.

The Wolf team from Novosibirsk, Russia, explored the selective writing of multi-core fiber gratings using femtosecond lasers and proposed a femtosecond laser grating writing system [48] as shown in Figure 4. The multi-core fiber is fixed by a specially polished glass sleeve, and the sleeve is moved to the beam-focusing region using a three-axis linear stage. The femtosecond laser is injected from above and focused on the designated position of the fiber. By moving the fiber using a high-precision gas-bearing translation stage, the refractive index of the core is changed point-by-point to form a fiber grating. Two CMOS cameras on the sides of the polished faces are used to precisely calibrate the position of the femtosecond laser writing on the transverse plane of the fiber and to monitor the entire writing process in real time. The team successfully prepared a fiber grating array on a twisted seven-core fiber using 1030 nm femtosecond lasers [49]. Subsequently, the team proposed to use this method to inscribe FBG on the SCF with spiral core and made side core FBG arrays, selected side core and center core FBG arrays, and a cross-sectional FBG array [49].

Wang Yiping’s research group at Shenzhen University used a femtosecond laser with a wavelength of 800 nm combined with a phase mask with a period of 1070 nm and a 0th-order diffraction below 4% to selectively etch the Bragg grating onto one of the cores in the dual-core dual-mode fiber [50]. The schematic diagram of the writing system is shown in Figure 5a. The dual-core dual-mode fiber consists of two identical cores, each of which is composed of a three-layer structure and can support two modes of transmission simultaneously. Since the writing light is focused on the position of core A, and the laser intensity at the position of core B is much lower than that of core A, an effective grating structure cannot be formed in core B. From the spectrum of core A shown in Figure 5a, three reflection peaks with wavelengths of 1546.08 nm, 1547.65 nm, and 1549.05 nm can be observed, which are generated by LP11 forward and backward mode coupling, LP01~LP11 cross-mode coupling, and LP01 forward and backward mode coupling, respectively. Chi Liu et al. used a nonlinear photoluminescence imaging technique of optical fibers to visually observe the position of femtosecond laser focus lines inside MCF [51]. This technique ensures that, when using a femtosecond laser with a phase mask method to inscribe FBG, the focused laser can be precisely positioned in the preset core, as shown in Figure 5b. By using this technique, the FBG with a different resonant wavelength of each core is inscribed in the same axial position of the SCF by pre-stretching the fiber. FBG has good consistency and a high edge rejection ratio, and the full width at half maxima of each FBG is less than 0.3 nm.

Wang’s team proposed a method for the femtosecond laser auto-positioning direct writing of a multi-core fiber Bragg grating array for shape sensing, which achieves automatic positioning of femtosecond laser point-by-point through image recognition and micro-displacement compensation [23]. This method does not require the removal of the coating layer or the rotation of the fiber during the grating writing process. Figure 6a shows the device for fabricating the FBG array using laser auto-positioning point-by-point technology. First, a top-down microscopic image of the fiber core is captured by a CCD, and then, the captured image is subjected to differential Gaussian filtering to increase the contrast between the fiber core and cladding boundary, effectively eliminating the adverse effects of illumination changes. The average gray value of each row of pixels is calculated to generate a digitized intensity profile of the fiber cross-section along the y-axis. Subsequently, an auto-correlation calculation is performed on the digitized intensity distribution of the fiber cross-section. The fiber is scanned vertically along the z-axis until the auto-correlation peak reaches its maximum value, thus finding the center of the core along the z-axis. The fabrication of the FBG array mainly consists of three steps. The first step is to fix the seven-core fiber on the processing platform and move the translation stage to find the initial position, i.e., core 1. Then, the fiber core is pre-scanned along the x-axis for a distance of 2 mm. This process employs an automatic positioning method for the fiber core, which simultaneously determines the three-dimensional trajectory for writing the fiber grating. Subsequently, the laser is turned on, focusing the femtosecond laser on the center of the core. During the translation of the multi-core fiber along the x-axis, the laser output is synchronized with the translation stage. The second step is to close the laser shutter after the first FBG is written, move the multi-core fiber along the y and z axes, and find the center of the next core based on the geometric shape of the seven-core fiber cross-section. Then, the shutter is reopened, and another FBG is written in the next fiber core. As shown in Figure 6b, the fabrication order of the seven-core fiber gratings is from bottom to top to ensure that the cores are not blocked. Step 1 is repeated with the same parameters to embed the other six FBGs sequentially into cores 2, 3, 0, 6, 5, and 4 of the SCF. In the third step, if the fabrication of the seven gratings in node 1 is completed, the shutter is closed, the fiber clamp is loosened, and then, the seven-core fiber is moved to the next node of the fiber by rotating the fiber spool through a stepper motor in the fiber winding system. Then, the fiber clamp is closed again, automatic positioning is used to find the fiber core, step 2 is repeated to fabricate the FBG in the second node, and this process is repeated to fabricate FBGs in the third to 20th nodes. Gratings with different grating periods are fabricated at each node, resulting in different wavelengths. This laser auto-positioning point-by-point technology for fabricating multi-core fiber FBG arrays is efficient, convenient, and feasible.

Different grating writing methods have their own characteristics. The full-core inscribing method has low cost and fast production speed, and the core-by-core writing method has high accuracy, but the cost is high. The characteristics of different processing methods are given in Table 1.

## 3. The Sensing Application of the MCF-Based FBG

With the gradual maturity of the preparation process of multi-core fiber gratings, various sensors based on multi-core fiber gratings have also emerged. The common core distribution of multi-core fibers is usually composed of one central core and several off-center cores. Therefore, the off-center cores are sensitive to bending, as bending of the fiber generates tangential strain at the off-center core positions. Bend sensitivity can be used in bend sensors to measure various physical parameters related to bending, such as acceleration, vibration, and fluid flow rate. Therefore, multi-core fibers have a wide range of applications in sensing directions such as bending, torsion, and shape detection.

### 3.1. Multi-Core Fiber Fan-In and Fan-Out Device

For high-density multi-core fibers, an important issue that needs to be addressed is how to insert and connect them to common single-mode fiber systems with low loss, whether it is for spectral quality monitoring during the grating writing process or for the application of multi-core fiber grating devices. To this end, a special fan-in/fan-out device (splitter) has been proposed and widely used in multi-core fiber applications. This device can independently transmit signals from each core of the multi-core fiber to a single-core fiber port. At present, there are three types of common fan-in and fan-out devices, including the fiber bundle, free space light method, and three-dimensional waveguide.

There are two common methods for optical fiber bundles, the bundling method and the intubation method. The bundling method uses hydrofluoric acid solution to etch part of the cladding of the single-mode fiber, so that the diameter of the cladding is equal to the new spacing of the multi-core fiber, and then inserts the etched single-mode fiber into two glass capillaries to form an optical fiber bundle. The capillary is fixed with a ferrule, polished on the end face, and finally fixed with an adhesive [52]. Thus, an optical fiber bundle with one end of the seven-core fiber size and the other end of the seven single-mode fibers as a fan-in/fan-out device is formed. This bundling method requires fine fiber processing and arrangement, and the operation process requires strict control. The fiber arrangement also requires high-precision equipment, so it is used less, but it can significantly reduce insertion loss and has low cross-talk. The other is the intubation method, which inserts multiple single-core fibers into a custom porous capillary according to the number of cores of the multi-core fiber and, then, heats and melts the arranged fiber bundle, similar to fiber tapering, to stretch the entire fiber bundle to the same outer diameter as the multi-core fiber, usually 125 μm. Then, the fiber bundle is cut in the middle of the cone to obtain an end-face structure that is the same as the multi-core fiber end-face [53,54]. This intubation method for taper preparation of fan-in/fan-out devices has a relatively simple operation process and requires less equipment, only requiring strict control of the tapering process. With the continuous improvement of processing technology, the insertion loss of fan-in/fan-out devices made by this method has gradually decreased, and the insertion loss and cross-talk can be controlled below 1 dB and −60 dB, respectively [55,56,57]. In addition, the fan-in/fan-out devices made by this method have a small size, are easy to package, and can be mass-produced. They are currently widely used fan-in and fan-out device processing methods.

The free-space optical method utilizes bulk optical components such as lenses, prisms, and adjustment mounts to adjust and optimize the coupling of multi-core fibers with multiple single-core fibers, forming a fixed optical path and fabricating multi-core fiber fan-in/fan-out devices [58,59]. Typically, a collimator is added to the front end of the single-mode fiber, and a single lens is used to focus the signal onto the input end of the multi-core fiber. This method allows for the individual adjustment of each core in the multi-core fiber with single-core fibers, resulting in low cross-talk. However, it requires high precision and stability of the adjustment mounts and optical components and has a relatively large volume. This method is suitable for applications with a small number of cores.

The three-dimensional waveguide type utilizes various substrates such as glass, polymers, silicon-based materials, and silicon nitride to transmit light from each core in a multi-core fiber to multiple single-core fibers through a waveguide structure. With the continuous advancement of femtosecond processing technology in recent years, using femtosecond direct writing to fabricate 3D waveguide fan-in/fan-out devices is also a possible method. By focusing the femtosecond laser beam, it can be directly inscribed inside the substrate material to form a 3D structure, realizing a channel connecting the multi-core fiber and multiple single-mode fibers [60,61]. The three-dimensional waveguide fan-in/fan-out device can be flexibly processed according to the structure of the multi-core fiber, with advantages such as one-time molding, high precision, and no limitation on the number of cores. However, the processing cost is higher, and the time is longer.

### 3.2. Sensors with MCF FBG

With the gradual maturity of MCF FBG preparation technology, sensing applications based on MCF FBG have also emerged. The common core distribution of multi-core fibers is mostly composed of one central core and several side cores deviated from the central core. Therefore, the side cores deviating from the central core are sensitive to bending, because the bending of the fiber will generate tangential strain at the position of the side core [62,63]. The core on the inner side of the bend will produce a compression response, while the core on the outer side will produce a tensile response. Therefore, by collecting the strain response values of each core, the local bending direction and bending radius of the fiber can be obtained. This characteristic makes multi-core fibers very suitable for applications in shape sensing and measuring various physical parameters related to bending, such as acceleration, vibration, and liquid flow rate [64,65,66,67].

Using multi-core fiber gratings as shape sensors for flexible instruments has advantages in various flexible medical devices, such as endoscopes and medical catheters. In limited space environments, a multi-core fiber is more space-efficient compared to multiple single-core fibers. Additionally, the spacing between the multiple cores in a multi-core fiber is fixed, and the temperature between the cores is the same. By analyzing changes in the reflection spectrum of the fiber grating, the impact of external physical shocks on the fiber can be detected. For example, compression and tension changes in the fiber can lead to changes in the effective refractive index and period of the fiber grating, which can result in spectral drift of the reflection peak, as shown in the following formula:(1)Δλ=pΔε+ΩΔT
where *p* and Ω are the bending strain sensitivity coefficient and the temperature sensitivity coefficient, respectively, Δ*ε* is the strain change, and Δ*T* is the temperature change. However, in the case of the MCF, the temperature of each core can be assumed constant. Since the FBG on the side core is not located on the central axis, it exhibits bending sensitivity.

We take SCF as an example to explain the bending sensing. In conditions of bending, SCF can be considered as an elastic beam with a circular cross-section. The schematic diagram for the cross-section of the SCF is shown in Figure 7. When the optical fiber undergoes bending, the relationship between the strain in each core and the bending curvature of the optical fiber is as follows:(2)$$\Delta \varepsilon_{ci}=D_{ci}\cdot \kappa =r_{i}\sin(\theta_{b}-\pi /2-\theta_{i})\cdot \kappa$$
where *i* = 1, 2, …, 6 represents the side core, *D*_ci_ is the distance from the side core c*i* to the neutral surface, *κ* is the curvature, *r_i_* is the distance from the center of the side core to the optical fiber geometric center, *θ_b_* is the bending direction angle, which represents the angle between the bending direction and the positive y-axis, and *θ_i_* is the angle between the core c*i* and the positive y-axis. The six side cores of the SCF are geometrically symmetrical. Consequently, when the optical fiber bends, two cores on the same axis exhibit one in stretching and the other in compression, meaning that the magnitude of their strain changes is the same, but the directions are opposite. Then, there is the following relationship: $\Delta \varepsilon_{c1}=-\Delta \varepsilon_{c4}$, $\Delta \varepsilon_{c2}=-\Delta \varepsilon_{c5}$ and $\Delta \varepsilon_{c3}=-\Delta \varepsilon_{c6}$. The six FBGs on the side cores can be considered as three sets of FBGs arranged in an equilateral triangular pattern, named as fc*i* (*i* = 1, 2, 3). The wavelength shifts of fc*i* can be represented as
(3)$$\left\{\begin{array}{ll} \Delta \lambda_{\mathrm{fc}1} & =\Delta \lambda_{1}-\Delta \lambda_{4}=p_{c1}\cdot \Delta \varepsilon_{1}-p_{c4}\cdot \Delta \varepsilon_{4}=2p_{c1}\cdot \Delta \varepsilon_{1}=2p_{c1}\cdot r_{1}\cdot \sin(\theta_{b}-\pi /2-\theta_{1})\cdot \kappa \\ & =p_{fc1}\cdot \kappa \\ \Delta \lambda_{\mathrm{fc}2} & =\Delta \lambda_{2}-\Delta \lambda_{5}=p_{c2}\cdot \Delta \varepsilon_{2}-p_{c5}\cdot \Delta \varepsilon_{5}=2p_{c2}\cdot \Delta \varepsilon_{2}=2p_{c2}\cdot r_{2}\cdot \sin(\theta_{b}-\pi /2-\theta_{2})\cdot \kappa \\ & =p_{fc2}\cdot \kappa \\ \Delta \lambda_{\mathrm{fc}3} & =\Delta \lambda_{3}-\Delta \lambda_{6}=p_{c3}\cdot \Delta \varepsilon_{3}-p_{c6}\cdot \Delta \varepsilon_{6}=2p_{c3}\cdot \Delta \varepsilon_{3}=2p_{c3}\cdot r_{3}\cdot \sin(\theta_{b}-\pi /2-\theta_{3})\cdot \kappa \\ & =p_{fc3}\cdot \kappa \end{array}\right.$$
where *p_fci_* is the bending sensitivity of fc*i*. It can be obtained through calibration. The bending direction and curvature of the optical fiber can be calculated by measuring the wavelength shift of FBGs and combining it with formula (3) [68].

#### 3.2.1. Single-Point MCF FBG Sensor

As early as 2000, M.J. Ander et al. researched multi-core fiber grating sensors [33]. By inscribing FBG on a four-core fiber (FCF) and selecting two diagonal fiber cores for signal monitoring, they realized a bending sensor with one-dimensional direction recognition and eliminated the crosstalk caused by axial strain and temperature. Subsequently, G. M. H. Flockhart et al. measured the two-dimensional bending direction through the FBG monitoring of three FBGs of the FCF [69]. Since then, researchers have carried out a lot of research on MCF FBG bending sensors.

G.A. Cranch et al. inscribed two groups of FBGs on an FCF to form a Fabry–Perot (F-P) interferometer for two-dimensional bending measurement [70,71]. The measuring system was demodulated by a Michelson interferometer, and the bending resolution of the sensor was obtained as ~ 0.012 km^−1^/Hz^1/2^. Kaiming Yang et al. aimed to achieve the measurement of one-dimensional bending by inscribing FBG on the dual-core few-mode fiber and using the reflection spectra coupled with different modes in the core [50]. DI ZHENG et al. proposed a demodulation scheme for MCF FBG bending sensor that uses a wideband sinusoidal wave source to convert the wavelength shift of the FBG into a change in reflected power so that the peak wavelength of the FBG can be accurately queried using a single photodiode [72]. The measurement error of the proposed demodulation method was within ±20 pm. This method was used to demodulate the direction and curvature of the MCF FBG sensor, and the curvature error measured by the test was less than 8%.

Maoxiang Hou et al. inscribed FBG on SCF, and the bending direction and curvature can be measured by using the spectrum of FBG with any two non-diagonal side cores, as shown in Figure 8a. The SCF has a total of 6 edge cores, so 12 combinations can be used to restore the curvature and azimuth of the bend. The results of multiple reversions can be averaged to obtain more accurate values. In a polar-coordinate system, the bend sensitivities display ‘8’-shaped patterns, which exhibit strong bending-direction dependence, as shown in Figure 8b. Experimental results show that the average relative error of measured curvature is less than 4.5%, and the average relative error of bending azimuth is less than 2.8% [43].

Hongye Wang et al. proposed a single-channel demodulated MCF FBG bending sensor to solve the complex problem of the MCF FBG demodulation system. The FBG with different reflection wavelengths of each core in the same axial position of FCF is inscribed by using ultraviolet core-by-core technology, and then, the coupling of FCF and SMF is realized by using thermal diffusion technology so that the signal of FCF FBG can be measured by SMF [73]. At the same time, the core symmetry of the FCF is utilized to enhance the bending sensitivity while eliminating the cross-talk of temperature and axial strain through differential calculation. Xingyong Li et al. also proposed a method of single-channel demodulation of MCF FBG signals. The cladding waveguide is constructed between the SMF and the FCF by femtosecond, and the coupling of the SMF and the FCF is realized [74].

Because of the unique bending sensing characteristics of MCF FBG, it is also used in the development of acceleration sensors. Amanda Fender et al. proposed an acceleration sensor based on FCF [64]. The sensor can measure acceleration in two vertical directions. When the frequency is lower than 300 Hz, the measurement error of acceleration is less than 5%, and when the temperature changes by 300 K, the standard deviation is less than 4 pm, so the sensor is temperature-insensitive. JINGXIAN CUI et al. proposed a two-dimensional accelerometer based on SCF FBG with a maximum resonance frequency of 149 Hz [75]. Vibration orientation can be obtained only by monitoring the wavelength displacement of the center core and two side cores that are not aligned in a straight line, and the accuracy range is 0.127°–2.888°. With this structure, both directional and acceleration information can be obtained in a single fiber.

Rui Zhou et al. designed and fabricated a three-dimensional accelerometer based on SCF FBG, where the fiber is installed in a metal structure, which can not only protect the sensor but also improve the sensitivity of the sensor to vibration signals [67]. The vibration azimuth and acceleration in the XY plane can be obtained by monitoring the wavelength of two side cores in the SCF, and the vibration acceleration in the z-axis can also be obtained by monitoring the wavelength displacement of the fixed core along the z-axis. Finally, the vector information of the vibration signal is obtained by vector synthesis. The designed accelerometer achieves an operating frequency bandwidth from 10 Hz to 220 Hz, with an optimal sensitivity of 355 pm/g and an azimuth accuracy of 0.269◦. Xunzhou Xiao et al. used the three side cores and the center core of SCF to realize vector acceleration measurement and eliminate the cross-talk of temperature [76].

Besides bending measurement, the SCF FBG sensor can also be used to measure other physical parameters. Wenbin Hu proposed a SCF FBG sensor that can measure both temperature and refractive index [77]. The cladding of SCF is corroded by the chemical corrosion method, so that the six side cores show a sensitive response to the change in surrounding refractive index, while the center core of SCF is insensitive to the refractive index but sensitive to temperature. The FBG of the center core can be used to realize the online temperature compensation of refractive index sensing. Javier Madrigal et al., for the first time, inscribed a thermal regeneration FBG on the MCF, enabling the measurement of multiple parameters under extreme environmental conditions [78]. The sensor can measure strain and curvature at 1000 °C. Yin Liu et al. achieved the simultaneous measurement of torsion and vector bending by inscribing FBG on a twisted FCF [79]. Hongye Wang et al. achieved the simultaneous measurement of bending, temperature, and pressure by combining MCF FBG with an F-P interferometer [80].

#### 3.2.2. Array MCF FBG Sensor

Based on the vector bending characteristics of MCF FBG, multiple MCF FBGs are cascaded, and then, linear interpolation, quadratic interpolation, and B-spline interpolation are used to make the discrete data continuous and finally realize the shape sensing of the whole fiber. In recent years, with the continuous development of MCF shape sensing technology, shape sensing based on MCF FBG has been used in medical, aerospace, civil engineering, and other fields.

Roger G. Duncan et al. proposed to use FBG sensor array to build a fiber-optic global positioning system for the shape positioning of search and rescue robots [81]. Three hundred FBG sensors (100 FBG sensors per core) were embedded in the shape-sensing cable composed of three core fibers about 1 m long, and then, experimental studies were carried out under the conditions of cantilever beam bending, three-point bending, and dynamic excitation. The experimental results show that the optical fiber positioning system built with the FBG sensor array can accurately reconstruct the two-dimensional and three-dimensional shape of the structure, with an error value of 1.2%. Lee et al. used a femtosecond laser to fabricate three waveguides and nine FBGs in a coreless fiber to achieve temperature-compensated deformation sensing measurements [82]. Three-dimensional stress and temperature monitoring can be realized at the same time. Under the sampling rate of 1 kHz signal, the detection bending radius can reach 40 mm, and the temperature range is 24 °C–100 °C. The current curvature sensing accuracy of the sensor can reach ±1.1 × 10^−3^ mm^−1^, and the position accuracy is 0.6 mm.

Paul S. Westbrook et al. achieved ultra-long-distance-distributed-strain sensing by inscribing continuous arrays of weak FBG in a twisting SCF [83,84]. To continuously inscribe the FBG, the fiber is coated with a transparent ultraviolet coating. In order to detect the twisting direction of the fiber, they also used twisted SCF. The shape reconstruction for the bending of different radii is shown in Figure 9. When external twisting occurs in the fiber, the torque length of the fiber is increased or reduced, so as to detect the magnitude and direction of the external torsion from the reflected signal. They used OFDR to monitor the reflected signal and reconstruct the reflected signal data. Cailing Fu et al. proposed a reflection shape sensor in the optical frequency domain based on a femtosecond laser inscribing weak FBG arrays in MCF [85]. An array of 60 weak FBGs was successfully inscribed on a 60 cm MCF, with each weak FBG having a length and spacing of 2 and 8 mm, respectively. The strain distribution of each core in two-dimensional and three-dimensional shape sensing is demodulated successfully by using the traditional cross-correlation algorithm. Using the apparent curvature vector method based on the Bishop framework, the minimum reconstruction errors per unit length of two-dimensional and three-dimensional shape sensors are increased to 1.08% and 1.07%, respectively.

In recent years, researchers have carried out research on the application of MCF FBG shape sensors in the medical field. Fouzia Khan et al. proposed a shape reconstruction technique using MCF FBG sensors as shape sensors for flexible medical devices [25]. Using the Frenet–Serret equation, the measured discrete MCF FBG data were continuous and the three-dimensional shape was reconstructed. The catheter, as shown in Figure 10, consisting of four MCFs, was placed in eight different configurations, and the reconstructed results were compared with the real values. The maximum reconstruction error in all configurations was 1.05 mm. Subsequently, the team proved the feasibility of using Bishop Frames to reconstruct the shape [86]. The experimental results showed that the average position error was less than 4.69 mm, and the average attitude error was less than 6.48°. Kangpeng Zhou et al. proposed a spiral MCF three-dimensional shape sensor for continuous surgical robots (FBG-Based 3D Shape Sensor Based on Spun Multi-Core Fibre for Continuum Surgical Robots) [87]. The Frenet–Serret frame was used to realize three kinds of shape reconstruction, and the sensing characteristics of torsion, strain, and temperature were systematically analyzed. The mean relative error of each shape was 0.49%, 1.90%, and 5.13%, respectively.

## 4. Summary and Perspectives

In recent years, the sensing applications of multi-core fiber Bragg grating devices have been increasingly emphasized. This article summarizes the main methods of the inscription of Bragg gratings in MCFs, including whole-core inscription and core-by-core inscription. It introduces in detail some applications of MCF FBGs in sensing, which can be categorized into single-point FBG and array FBGs based on the number of gratings in MCF. For single-point FBG, bend sensing is the main physical quantity measured, which can further develop into acceleration sensors. Furthermore, by constructing different types of fiber interferometers in MCFs and combining them with MCF FBG, a variety of sensor measurements for different parameters can be achieved. Using the array FBG in MCF can better perform shape sensing, and by signal demodulation algorithms, the actual shape of MCF FBGs can be reconstructed.

In the fabrication methods of MCF FBGs, each writing technique has its own advantages. The full-core inscribing method typically does not demand high stability from the processing system, and it achieves a high success rate in FBG inscription. However, when inscribing FBGs with different periods or types on a same fiber, the need to change phase masks reduces flexibility. The shadowing effect of the cores cannot be entirely avoided, leading to a lack of uniformity in the FBGs of each core. The core-by-core writing method can control the writing process, which makes it more flexible and convenient. And it can mitigate the lens effect and shadow effect inherent to the fiber itself. However, due to the physical size limitations of the fiber, this method requires high precision in motor control, increasing the difficulty of the writing process. Future research work will see continuous improvements in the fabrication schemes for MCF FBGs with the development and application of inscribing equipment, inscribing technology, and new materials.

In terms of sensing applications, compared to traditional electrical sensing, MCF FBGs have an advantage in bending sensing due to the characteristic distribution of the cores within the MCF. By constructing an array of FBGs and using demodulation algorithms, shape reconstruction can be achieved, making them extremely useful in shape sensing applications. Interest in shape sensing with MCF FBGs is growing, and demands across different industries are driving the development of MCF FBG shape sensing research. This mainly includes monitoring structural deformations in extreme environments, real-time tracking of catheter shapes in medical applications, precise navigation of surgical robots, real-time posture sensing of robots, etc. Currently, the application of shape sensing with MCF FBGs still faces several challenges, mainly including the following: 1) There is a lack of unified industrial standards for MCF and its devices and poor compatibility between different devices, making it difficult to promote industrial mass production and reduce costs; 2) The core spacing of MCF currently in use is relatively small, and there is still room for improvement in accuracy for fiber shape sensors with larger core spacing. As the application fields expand and research continues to improve, the sensing performance of MCF FBGs will be further enhanced.

## Figures and Tables

**Figure 1 sensors-24-04532-f001:**
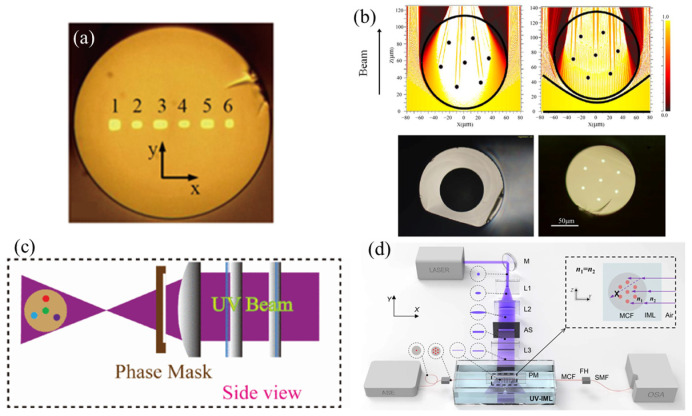
(**a**) Photo of the cross-section of the heterogeneous multi-core fiber [35] (**b**) using a polished capillary to counteract the shadow effect [37], (**c**) phase mask method based on defocus writing [39], (**d**) phase mask method based on external ultraviolet transparent media [40].

**Figure 2 sensors-24-04532-f002:**
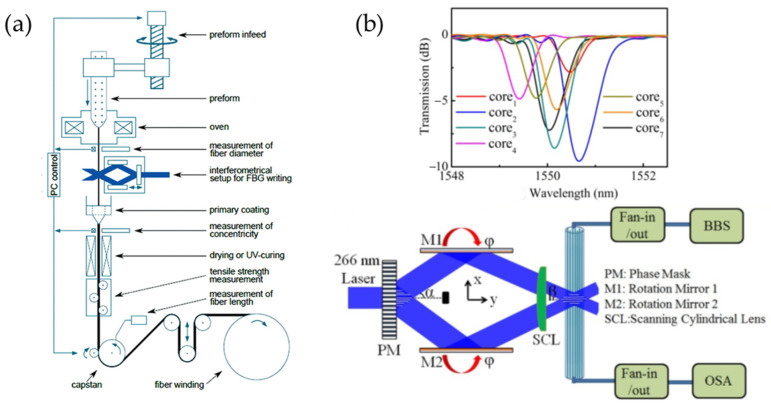
(**a**) Schematic diagram of DTG [42], (**b**) the transmission spectra of the FBGs inscribed by modified Talbot interferometer, and the schematic diagram of the modified Talbot interferometer [43].

**Figure 3 sensors-24-04532-f003:**
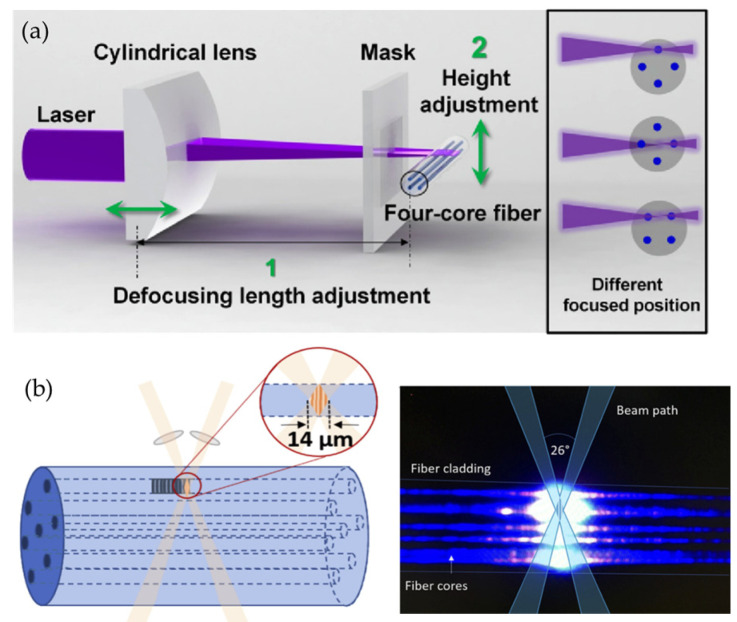
(**a**) Core-by-core writing based on phase mask [44], (**b**) small spot ultraviolet direct writing method [45].

**Figure 4 sensors-24-04532-f004:**
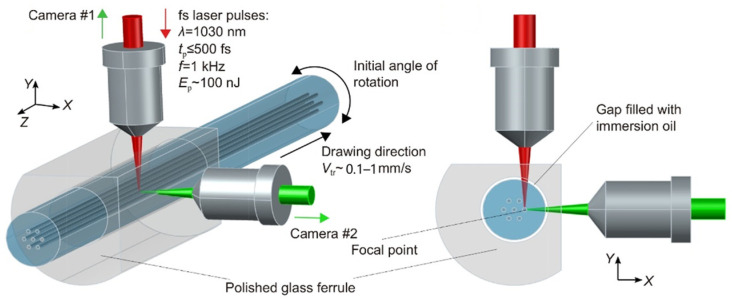
Femtosecond laser direct writing FBGs in seven-core fiber [48].

**Figure 5 sensors-24-04532-f005:**
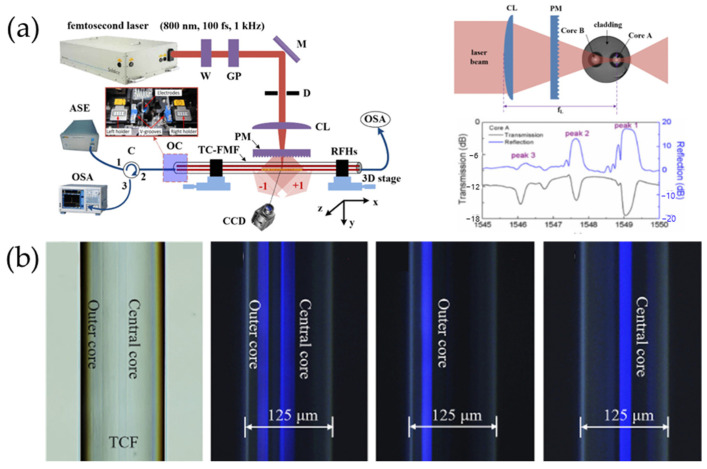
(**a**) FBG writing system combining femtosecond laser and phase mask [50], (**b**) selective inscription method by femtosecond laser and phase mask [51].

**Figure 6 sensors-24-04532-f006:**
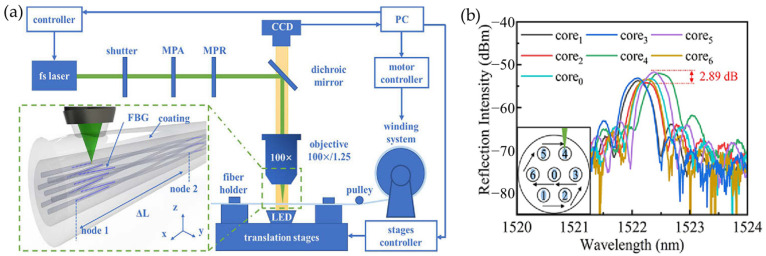
(**a**) Inscribing array FBG by auto-positioning point-by-point technology, (**b**) fabrication order of the seven-core fiber [24].

**Figure 7 sensors-24-04532-f007:**
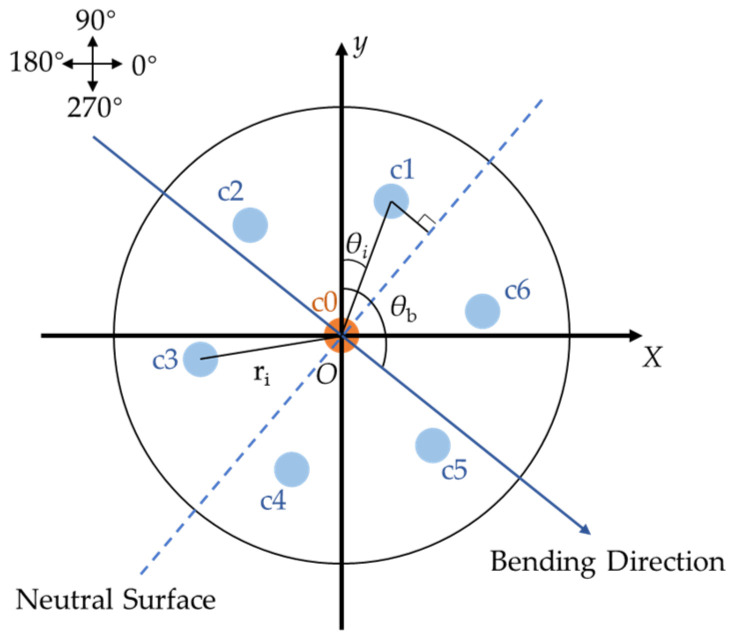
Schematic diagram for the cross-section of the SCF, in which the center core is colored orange and the side core is colored light blue.

**Figure 8 sensors-24-04532-f008:**
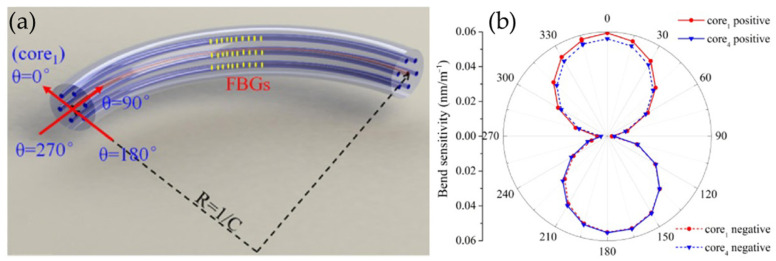
(**a**) The two-dimensional bending of the seven-core FBGs, (**b**) bend sensitivities for two FBGs [43].

**Figure 9 sensors-24-04532-f009:**
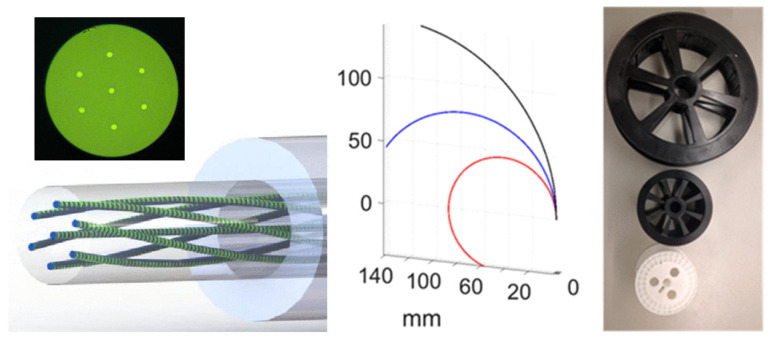
The twisted seven-core fiber schematic and shape reconstruction for bending of different radii, the red, blue, and black lines correspond to the shape reconstruction of the spools with a radii of 4.5cm, 7.62cm, and 14.6cm, respectively. [83].

**Figure 10 sensors-24-04532-f010:**
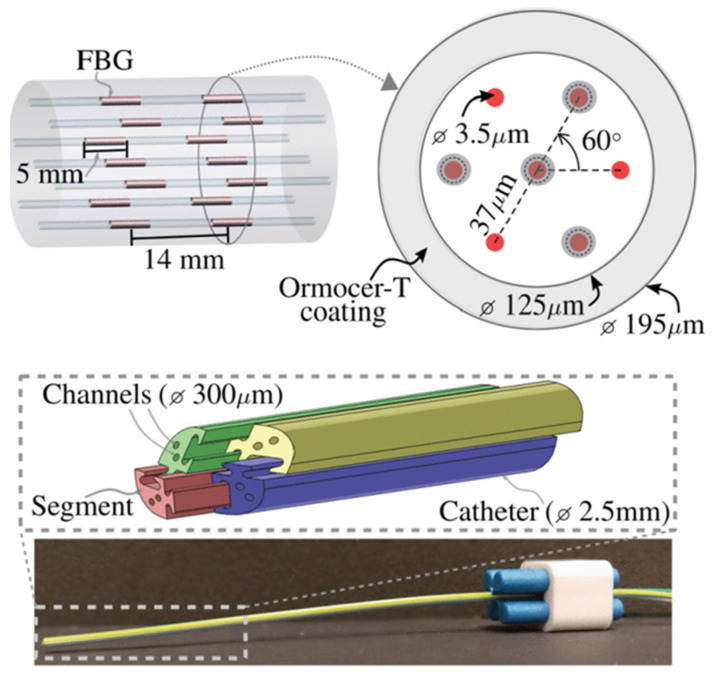
The four-segment catheter used for pose measurement [86].

**Table 1 sensors-24-04532-t001:** The features of the methods of inscribing FBG-MCF.

Reference	Type of Optical Fiber	Method of Inscribing FBG	Features
[33,34]	Four-core fiber with square core arrangement	Phase mask method based on UV laser inscribing of full-core	Low production cost, fast production speed, poor reflectivity consistency
[36,37,38,40]	Seven-core fiber	Phase mask method based on UV laser inscribing of full-core	Low production cost, fast production speed, poor reflectivity consistency
[39]	Four-core fiber with triangular core arrangement	Phase mask method based on UV laser inscribing of full-core	Low production cost, fast production speed, poor reflectivity consistency
[44]	Four-core fiber with square core arrangement	Phase mask method based on UV laser inscribing of core by core	High controllability, low production cost, complex production process
[45]	Seven-core fiber	Double-beam interference based on UV laser inscribing of core by core	High controllability, low production cost, complex production process
[24,48,49]	Seven-core fiber	Point-by-point inscribing method based on femtosecond laser	High controllability, high production cost
[50]	Dual-core fiber	Phase mask method based on femtosecond laser inscribing of core by core	High controllability, high production cost, difficulty focusing
[51]	Seven-core fiber	Phase mask method based on femtosecond laser inscribing of core by core	High controllability, high production cost, difficulty focusing

## Data Availability

Data sharing is not applicable.

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
