# Peer review of "Multi-Core Fiber Bragg Grating and Its Sensing Application"

_sensors, 2024, doi:10.3390/s24144532_

Round 1

Reviewer 1 Report

Comments and Suggestions for Authors

The authors reviewed the primary methods for inscribing Bragg gratings in microchannel optical fibers, including full-core and core-by-core inscription. It thoroughly explored the applications of fiber Bragg gratings in the sensing field, categorizing them into single-point and array fiber Bragg gratings based on the number of gratings. Specific applications include bend sensing, acceleration sensing, shape sensing, refractive index sensing, temperature sensing, and strain sensing. Several suggestions are supplied:

1.      It is recommended that the author correct the figure number in line 195.

2.      The classification of femtosecond laser inscription methods for FBGs is not rigorous enough. It is recommended to include the femtosecond holographic interference inscription method.

3.      The classification of inscription methods into full-core and core-by-core based on the number of cores inscribed in MCF requires further justification to establish its necessity.

4.      In the summary section, the discussion on the challenges faced by MCF FBG shape sensing applications in the third point has logical issues. It is recommended to reorganize this section to improve coherence and clarity.

5.      The conclusion of the article lacks depth. The discussion on the current issues with FBG inscription techniques and the potential future solutions is neither detailed nor profound enough.

6.      The references cited in the article are not typical and are inadequate in number.

Comments on the Quality of English Language

Moderate editing of English language required

Author Response

Please see the attachment。

Reviewer 2 Report

Comments and Suggestions for Authors

Comment on “Multi-core Fiber Bragg Grating and its Sensing Application”

Zhang et al reviewed the methods of inscribing and sensing application for multi-core fiber Bragg grating (MCF-FBG). The content in this paper is interesting because the research on MCF-FBG is still new and from this paper will help as a guidance for future development. Therefore, I accept these articles with major revision and hope that the authors can provide reasons for the paste preparation. Below are my comments and suggestions.

      In Section 2 and Section 3, it is recommended for the authors to provide the table of summarize for methods of inscribe and sensing application of FBG-MCF to ensure the readers easy to compare the methods or performance for each previous development. You may refer the following articles:

Campanella, C.E.; Cuccovillo, A.; Campanella, C.; Yurt, A.; Passaro, V.M.N. Fibre Bragg Grating Based Strain Sensors: Review of Technology and Applications. Sensors 201818, 3115. https://doi.org/10.3390/s18093115  

      Is there any mathematical modelling or theory on the FBG-MCF? Because it will provide more deep understanding to the readers on how the light propagates inside FBG-MCF and their characteristic.
